# COVID-19 and Obesity: Dangerous Liaisons

**DOI:** 10.3390/jcm9082511

**Published:** 2020-08-04

**Authors:** Grazia Caci, Adriana Albini, Mario Malerba, Douglas M. Noonan, Patrizia Pochetti, Riccardo Polosa

**Affiliations:** 1Internal Medicine, Luzerner Kantonsspital, LUKS, 6000 Luzern 16, Switzerland; grazia.caci15@gmail.com; 2Scientific and Technology Pole, IRCCS MultiMedica, 20138 Milan, Italy; albini.adriana@gmail.com (A.A.); douglas.noonan@uninsubria.it (D.M.N.); 3Traslational Medicine Department, Eastern Piedmont University (UPO), 13100 Novara, Italy; mario.malerba@uniupo.it; 4Respiratory Unit, St. Andrea Hospital, 13100 Vercelli, Italy; patrizia.pochetti@tin.it; 5Center of Excellence for the acceleration of HArm Reduction (CoEHAR), University of Catania, 95124 Catania, Italy; 6Department of Biotechnology and Life Sciences, University of Insubria, 21100 Varese, Italy; 7Dipartimento di Medicina Clinica e Sperimentale, University of Catania, 95124 Catania, Italy

**Keywords:** obesity, ACE2, renin–angiotensin system (RAS) pathway, COVID-19, SARS-CoV-2, cytokine storm, adipokines, IL-6 pathway

## Abstract

Besides advanced age and the presence of multiple comorbidities as major contributors to increased risk of severe disease and fatal outcome from Severe Acute Respiratory Syndrome coronavirus 2 (SARS-CoV-2) disease (COVID-19), there is now emerging evidence that overweight and obesity predispose to severe symptoms and negative prognosis. Remarkably, the severity of COVID-19 appears to rise with increasing body mass index (BMI). The association between COVID-19 outcomes and overweight/obesity has biological and physiological plausibility. Potential pathophysiological mechanisms that may explain this strong association include the chronic pro-inflammatory state, the excessive oxidative stress response, and the impaired immunity that is commonly reported in these individuals. The role of cytokines, mammalian target of rapamycin (mTOR), and altered natural killer cell polarization in the dangerous liaison between COVID-19 and obesity are discussed here. These pathways can favor and accelerate the deleterious downstream cellular effects of SARS-CoV-2. Moreover, obesity is well known to be associated with reduced lung function and poor response to mechanical ventilation, thus placing these individuals at high risk of severe illness and mortality from COVID-19. Furthermore, obesity may lead to other complications, such as renal failure, cardiovascular dysfunction, hypertension, and vascular damage, which in turn can further accelerate negative clinical outcomes from COVID-19. Obese individuals should be shielded against any potential viral exposure to SARS-CoV-2 with consequential considerations for compulsory protection devices and social distancing. Health care providers should be aware that obesity predisposes to severe symptoms and negative prognosis in COVID-19 patients.

## 1. Introduction

The initial “official” cases of the coronavirus disease 2019 (COVID-19) were notified in the city of Wuhan in Hubei Province, China, in December 2019 [1]. The outbreak was linked to a novel beta coronavirus related to the Severe Acute Respiratory Syndrome (SARS) virus [2]. As of the 31st of July 2020, globally, more than 17 million cases of SARS coronavirus-2 (SARS-CoV-2) had been confirmed, and 669,231 COVID-19-related fatalities had occurred (https://coronavirus.jhu.edu/map.html). Advanced age, male sex, and the presence of multiple comorbidities have been clearly identified as major risk factors for the development of severe COVID-19 [3,4,5], whereas counter-intuitive evidence shows reduced COVID-19 hospitalizations among smokers [6,7,8].

There is now emerging evidence that COVID-19 and its severity is associated with overweight and obesity. This is not unexpected. A significant association between obesity and disease severity and mortality was already reported for other respiratory virus pandemics, including that of 2009 H1N1 influenza [9,10].

In California, of the 268 adults hospitalized for 2009 influenza A (H1N1), 58% were obese (body mass index (BMI) ≥30), with morbid obesity (BMI ≥ 40) being most commonly associated with disease fatality [9]. A subsequent Mexican study not only confirmed these observations that influenza-positive adults were more likely to be hospitalized if obese (particularly if morbidly obese) but expanded these findings also to include many other viral pathogens (including coronavirus, metapneumovirus, parainfluenza, and rhinovirus) [10]. The Centers for Disease Control and Prevention considers people with BMI ≥ 40 kg/m^2^ as being at risk for flu complications [11]. Higher BMI is associated with increased health care expenditures due to increased use of hospital and health care services in the last 6 months of life [12]. Obesity patients can suffer more frequently from cardiovascular dysfunction and hypertension, and many obese patients have type 2 diabetes. Consequently, obesity may be an important risk factor for severe SARS-CoV-2 illness.

The emerging association between COVID-19 outcomes and overweight/obesity has biological and physiological plausibility. In overweight and obesity, macronutrient excess in the adipose tissues stimulates adipocytes to release tumor necrosis factor α (TNF-α), interleukin 6 (IL-6), and other pro-inflammatory mediators and to reduce production of the anti-inflammatory adiponectin, thus predisposing to a pro-inflammatory state and oxidative stress [13]. Moreover, obesity itself has been shown to impair immune responses with an overall negative impact on the efficiency of pathogen defenses [14]. Considering that obese people are predisposed to a pro-inflammatory state, oxidative stress, and impaired immunity, the acceleration of viral inflammatory responses in COVID-19 and most unfavorable COVID-19 prognoses are most likely to occur in these individuals. Moreover, the well-known association of obesity with reduced lung function and poor response to mechanical ventilation places people who are obese at risk of severe illness and mortality from COVID-19.

Here we will review the evidence suggesting that overweight/obese people are most vulnerable to COVID-19 and discuss potential pathophysiological mechanisms.

## 2. Epidemiological Data

Although the impact of overweight and obesity on COVID-19 outcomes has not yet been well described, the previous H1N1 influenza experience has clearly illustrated that patients with severe obesity were the most at risk for disease progression and mortality from influenza complications [9,10]. COVID-19 is no exception.

In a Chinese case series of 221 patients, severe illness of COVID-19 was independently associated with body mass index (BMI) ≥ 28 kg/m^2^ (odds ratio (OR), 5.872; 95% confidence interval (CI), 1.595 to 21.621; *p* = 0.008) [15]. In the USA, the prevalence of obesity is high and rising, with a high burden of class III obesity (9.2% of the population with BMI > 40 kg/m^2^) [11]. Thus, countries with a high prevalence of obesity and fast-growing COVID-19 cases will face unprecedented challenges. Though patients aged <60 years are generally considered a lower-risk group of COVID-19 severity, obesity appeared to be a previously unrecognized risk factor for hospital admission and need for critical care, independently of age. A retrospective analysis of BMI in USA SARS-CoV-2 patients, revealed that subjects aged <60 years with a BMI between 30 and 34 were 2.0 (95% 1.6–2.6, *p* < 0.0001) and 1.8 (95% CI 1.2–2.7, *p* = 0.006) times more likely to be admitted to acute and critical care, respectively, compared to individuals with a BMI < 30, while patients with a BMI > 35 and aged <60 years were 2.2 (95% CI 1.7–2.9, *p* < 0.0001) and 3.6 (95% CI 2.5–5.3, *p* ≤ 0.0001) times more likely to be admitted to acute and critical care compared to same-aged patients with BMI < 30 [16]. Another USA study has shown that COVID-19 deaths were more frequently associated with obesity (OR 3.1; 95% CI: 1.5–6.6), with morbid obesity showing the highest level of association (OR 7.6; 95% CI 2.1–27.9) even in patients with no other comorbidities [17]. Severe obesity (BMI > 35) has been identified as a major risk factor for SARS-CoV-2 infection in patients hospitalized with COVID-19 in the New York City area [18]. In another NY study, a total of 5700 patients were included (median age 63 years) [19]. The most common comorbidities were hypertension (56.6%), obesity (41.7%), and diabetes (41.7%); those with diabetes were more likely to have received invasive mechanical ventilation or care in the intensive care unit (ICU) [19]. During hospitalization, 12.2% of patients received invasive mechanical ventilation (IMV), 3.2% were treated with kidney replacement therapy, and 21% died. Mortality for those requiring IMV was 88.1% [19]. Additional large retrospective case series from New York confirm that obesity is a major risk factor for COVID-19 disease severity and intensive care unit requirements [20,21]. In a large prospective cohort of 502,543 middle-aged adults in the U.K., BMI and waist circumference were independently associated with laboratory-confirmed COVID-19 in a dose-dependent fashion [22]. Adjustment for possible confounders did not change the results. The adjusted odds ratio for overweight, obese, and severely obese subjects was 1.31 (95% CI: 1.05–1.62), 1.55(1.19–2.02), and 1.57 (1.14–2.17), respectively, compared to those with normal weight. In a retrospective French cohort study, the proportion of patients who required IMV increased with BMI categories (*p* < 0.01, chi-square test for trend), and it was greatest in patients with BMI > 35 kg/m^2^ (85.7%) [23]. In multivariate logistic regression, the need for IMV was significantly associated with male sex (*p* < 0.05) and BMI (*p* < 0.05), independent of age, diabetes, and hypertension. The odds ratio for IMV in patients with BMI > 35 kg/m^2^ vs. that for patients with BMI < 25 kg/m^2^ was 7.36 (1.63–33.14; *p* = 0.02). In a study in England of 10,926 COVID-19-related deaths, obesity (BMI > 40) had a hazard ratio of 1.92 (95% CI 1.72–2.13) [24]. Obesity represents the strongest predictor for COVID-19 followed by diabetes and hypertension in both sexes and chronic renal failure in females only, in a case–control study in Mexico [25]. In Italian patients who died of COVID-19, the comorbidity with hypertension was 67%, that with type 2 diabetes was 30%, and that for obesity was 11% [26], but their median age was 80 years.

In a small case series from Italy, Germany, and China: Obesity was present in 31% of patients, with an additional 58% being overweight among critically ill patients in Italy [27]. In Germany, patients with acute respiratory distress syndrome (ARDS) were more commonly overweight or obese (83%) versus those with normal BMI (42%) [28]. In China, patients with overweight/obesity were more likely to be hospitalized longer than those with normal BMI [29]. Therefore, obesity may be associated to COVID-19 morbidity and mortality independently of older age and presence of comorbidities. That obesity is emerging as a risk factor for COVID-19 outcomes regardless of age and comorbidity [30,31,32] is indication that even young people are at risk of serious disease if their BMI is elevated. In a USA study that had children and adolescents hospitalized with COVID-19 obesity was significantly associated with disease severity [33]. A retrospective case–control study of young Chinese patients with COVID-19 showed that obesity was the most important critical factor contributing to their death [34].

When interpreting the potential role of obesity as a risk factor for morbidity and mortality from COVID-19, the role of cardiovascular disease, hypertension, and diabetes cannot be discounted given that obesity is intimately related to these conditions, and all of them have been reported to be associated with a more severe case of COVID-19. One of the main medium-term consequences of obesity, considered as one of the most important risk factors that predispose to cardiovascular diseases in adulthood, is represented by increases in blood pressure values. Obesity would induce hyperinsulinism first and insulin resistance afterward, followed by type 2 diabetes mellitus.

## 3. Involved Pathways

The extracellular domain of angiotensin-converting enzyme-2 (ACE2) has been initially identified as a receptor for the spike (S) protein of SARS-CoV [35], and more recently also for that of SARS-CoV-2. SARS-CoV-2 cell entry depends on ACE2 and TMPRSS2 (an androgen-induced transmembrane protease) [36,37,38,39]. Many studies are focused on the correlation between obesity and the renin–angiotensin system (RAS), consisting of a series of enzymatic reactions that lead to the generation of different angiotensin peptides [40,41,42]. The renin induces proteolytic cleavage of angiotensinogen into angiotensin I, which is then converted into angiotensin II by the action of angiotensin-converting enzyme (ACE), which will then act on two angiotensin type 1 (AT1R) and type 2 (AT2R) receptors. However, two types of ACEs, namely, ACE1 and ACE2 are known. In 2000, *ACE2*, a homolog of the *angiotensin-converting enzyme* (*ACE*) gene, the encoded protein was reported to negatively regulate RAS by converting angiotensin (Ang)-II to Ang-1–7 [40,41]. The action of ACE1 will induce vasoconstriction, proliferation and inflammation, and fibrosis, while the action of ACE2 through activation of Mas receptors induces dilation, acts with anti-inflammatory and anti-fibrotic action, and even proves to be protective against sepsis [43,44,45].

ACE2 is widely distributed in our organism [42], and it is also expressed in adipocytes; its expression is upregulated in response to high fat diet induced obesity in mice [43]. The ACE2/Ang 1–7/MasR axis is a negative regulator of the ACE/Ang II/AT1 receptor (AT1R) axis. Angiotensinogen is expressed and constitutively secreted from adipose tissues by mature adipocytes in both humans and experimental animal models [46]. The angiotensin type 1 (AT1R) and type 2 (AT2R) receptors may mediate the effect of Ang II and cause upregulation of adipose tissue lipogenesis (mediated via AT2R) and downregulation of lipolysis (mediated via AT1R) [47]. Hypertension in obese patients dysregulated the RAS pathways [42,48].

Obesity induces a state of moderate chronic inflammation (Figure 1), IL-6 and TNF-α are consistently increased in circulation of obese humans and mouse models [49,50]. It induces an inflammatory state through the increase of macrophage infiltration into adipose tissue, the polarization state of macrophages and the augmentation of cytokines and chemokines [51,52,53]. IL-6 and STAT3 signaling act as critical determinants for obese models [49,50], and form a feed-forward loop. IL-6 and TNF-α serum levels remained independent and significant predictors of disease severity and deaths of COVID-19 patients [54,55].

Angiotensinogen and ACE1 are increased in obese individuals and mice [50], but ACE2 is not. Different studies pointed out the role of angiotensin (Ang)-1–7 in metabolic regulation, because Ang-1–7 seems to play an important role in anti-obesity [43]. ACE2 exerts an anti-obesity effect [56] through production of Ang-(1–7) and Mas receptor [57]. Ang-(1–7) is known to have an anti-obesity effect: 1) transgenic rats with high Ang-(1–7) levels are leaner and did not develop diet-induced obesity [57], 2) in mice, Ang-(1–7) given orally lowers body weight and fat mass [58,59], 3) in Mas-deficient mice, dyslipidemia, abdominal fat mass was increased, with glucose intolerance and reduced insulin sensitivity [45]. Knockout mice for *ACE2* have increase insulin-resistance and inflammation and induce a pro-inflammatory phenotype in macrophages and lung pathology [40,41].

Natural killer (NK) cells are effector lymphocytes of the innate immune system, members of the ILC family (innate lymphoid cells), and are essential for the elimination of cells that express low or undetectable levels of MHC I (major histocompatibility complex) or that have a high expression of stress ligands, both conditions associated with viral infections [60,61]. Patients with obesity show a significant reduction in the cytotoxic activity of NK cells [51,52,53], which is also associated in humans and animal models, with increased susceptibility to viral infections [51,52,53]. Overweight and obesity patients have reduced ADCC (antibody-dependent cellular cytotoxicity) in NK cells in vitro [62]. COVID-19 patients, in particular those that required intensive care (ICU), showed decreased numbers of circulating T CD4+, T CD8+ but also displayed reduced antiviral cytokine production capability [63]. These ICU patients also showed increased serum IL-6 levels, which correlated to the decreased frequency of granzyme-expressing NK cells and a reduced cytotoxic potential. Off-label treatment with tocilizumab, an anti-IL-6 monoclonal antibody, restored the cytotoxic potential of NK cells [63].

Obesity has been linked to a reduction in the diversity of T-cell receptors and also been shown to cause a reduction in lymph node size, inhibition of the number of T-cells in the lymph nodes, and reduced ability of the immune system to recognize and effectively deal with foreign antigens [64]. The expansion of adipocytes caused by obesity suppresses anti-inflammatory pathways, and constant presentation of antigens by DCs may eventually lead to T-cell exhaustion and chronic inflammation [64].

Obese people have also a higher level of leptin, which is a pro-inflammatory adipokine, and a lower concentration of adiponectin, which is an anti-inflammatory adipokine. This dysregulation can play a role also in immunomodulation, and it could explain the contribution to complications in SARS-CoV-2 patients [23].

The mammalian target of rapamycin (mTOR) is a conserved serine/threonine kinase that plays key regulatory roles in several biological processes, such as proliferation and survival of normal and malignant cells, differentiation, metabolism, and autophagy [65]. mTOR acts as a sensor of various environmental or cellular alterations, such as nutrient availability, energy status, and stress response, and stimuli, such as cytokines and hormones. Two distinct mTOR complexes exist: mTORC1 and mTORC2 [65]. mTORC1 drives multiple anabolic pathways, including protein and nucleotide synthesis and ribosome production. It releases the eukaryotic initiation factor eIF4E from its inhibitors by acting on the 4E-BP (eIF4E-binding protein), and it phosphorylates and activates ribosomal S6 kinase 1 (S6K1) [65]. Obesity upregulates the chronic hyperactivation of mTOR activity in multiple tissues, and increases S6K activity and over phosphorylation of translation suppressor 4E-BP [66]. Most coronavirus mRNAs are believed to undergo cap-dependent translation using eIF4F [66]. According to the hypothesis of Bolourian and Mojtahedi, in obese patients, hyperactivated mTOR regulates cap-dependent mRNA translation and coronaviruses are RNA viruses, hijacking the host cap-dependent translation machinery to replicate [66].

## 4. How Obesity Can Be Related to SARS-CoV-2 Disease Pathophysiology and Severity

Obesity is a condition that can be a result of an excess of food intake or altered energy expenditure; it is also a condition that is implicated in many other diseases such as atherosclerosis, hepatic steatosis (non-alcoholic fatty liver disease (NAFLD) a precursor to NASH, non-alcoholic steatohepatitis), and metabolic syndrome, with increased insulin resistance that could evolve into type 2 diabetes mellitus. Obesity should be considered not just a risk factor but a disease, with COVID-19 as a factor in this cause [67].

Given the extremely high rates of obesity around the globe, it is expected that a high percentage of the population infected with coronavirus will also have a BMI over 25. Furthermore, persons with obesity who become ill and require intensive care present challenges in patient management. In fact, obese patients have been shown to have more fatty tissue in the areas surrounding the larynx and pharynx segments that may undergo to compression, and it makes intubation more difficult. Obese patients are very prone to diminished airway flow, due to limited truncal expansion, making it challenging to ease the airflow [68]. In addition, they are more difficult to position and transport by nursing staff; like pregnant patients in intensive care units, they may not do well when prone, so special beds and positioning/transport equipment that are available mostly in specialized bariatric surgery units may not be widely available elsewhere in hospitals [69]. The COVID pandemic is rapidly spreading worldwide, especially in Europe and North America where obesity is highly prevalent, thereby increasing the risk of severe SARS-CoV-2 pneumonia requiring advanced respiratory support [19]. Considering that COVID-19 patients frequently require intensive care and mechanical ventilation and that obesity per se is a predisposer for mechanical ventilation in the ICU, it is not surprising to observe such a strong association of severe COVID-19 in patients who are obese. In patients with acute respiratory distress syndrome (ARDS), obesity is associated with a decrease in mortality [70]: COVID-19 seems to be challenging this “obesity paradox” [71].

After entering the cells, the SARS virus induces a systemic downregulation of ACE2 [72] to prevent another viral infection of the cell. As the cardiovascular and anti-inflammatory protective factor of the ACE2/Ang-1–7/MasR axis is lacking, the balance shifts even more to the pro-inflammatory side [42], also explaining the cytokine storm that follows, which is a synergistic effect of the physiological condition of adipose tissue and pathological virus-induced reduction of ACE2 [73]. The higher expression of ACE2 in the adipose tissue proves to be a positive counter-regulator of the damage induced by ACE1, on the other hand it favors viral entrance via the receptor bond mentioned above with reference to SARS-CoV-2. We should also consider the fact that obese patients, as already seen in H1N1 influenza, have an increased duration of virus shedding; symptomatic patients with obesity shed the virus for a 42% longer time than adults who do not have obesity [74]. This could be another point to note regarding the duration of the quarantine in these patients. Subjects with obesity affected by COVID-19 require longer hospitalization, more intensive and longer oxygen treatment, and they may have longer SARS-CoV-2 shedding [75]. This is probably due to the ACE2 on the adipose tissue in obese patients and because mTOR is activated in the adipose tissue.

Obesity is known to cause chronic inflammation and an increase in circulating, pro-inflammatory cytokines, which may play a role in the worsening of COVID-19 outcomes. IL-6 for example is elevated both in obesity patients [49,50] and in the ICU patients affected with SARS-CoV-2 [63]. Both SARS-CoV-2 infection and obesity seem to share some common metabolic and inflammatory reaction pathways [68]. Having a higher BMI not only increases the risk of infection and of complications for a single obese person, but recent evidence indicates that a large obese population increases the chance of the appearance of a more virulent viral strain, prolongs virus shedding throughout the total population, and eventually may increase the overall mortality rate of an influenza pandemic [76]. In particular, in this context, the acronym “MicroCLOTS” (microvascular COVID-19 lung vessels obstructive thromboinflammatory syndrome) also takes hold to describe a progressive pulmonary endothelial syndrome with microvascular thrombosis, which also explains the activation of the factors involved in coagulation (increase of D-Dimer) [33,55,77].

## 5. What Are the Implications of the Findings?

Given the magnitude of the effect, should people consider dieting until the pandemic subsides? Of course, such a highly provocative proposition is unpractical. Nevertheless, in the quest for new therapies to stem the spread of the coronavirus pandemic and limit morbidity and mortality from COVID-19, these findings may lead to different conclusions. There is well-known evidence of the beneficial role of chronic physical exercise in disease prevention, as adjuvant treatment in chronic diseases, and in psychological well-being [78]. Moreover, regular moderate exercise is associated with a reduced incidence of infection compared with a completely sedentary state [79]. Conversely, a single, acute bout of prolonged, strenuous exercise has a temporary depressive effect on immune function [79], so in an acute setting, physical exercise in obese patients should be planned only if integrated in a pulmonary rehabilitation program after COVID treatment and discharge.

Obesity is also a marker of poverty and, therefore, inability to access to high-quality health care [80]. The physiological consequences of obesity, such as inflammation, contribute to COVID-19 severity in people with high BMI. Obesity is a signal for social determinants of health, and it should be considered among health-disparity situations and a disease per se, with analysis of data related also to race, financial status, education, and access to health care and prevention [80].

## 6. Conclusions

Recent studies have shown that there is a high frequency of obesity among COVID patients admitted in intensive care units, the severity of SARS-CoV-2 disease increases with BMI, and severe obesity may lead to other complications such as heart failure and renal failure and lead to renal support and advanced respiratory care. This hypothesis is based on an accessible summary of the epidemiological evidence and the pathophysiological mechanisms regarding obesity and COVID-19. These results suggest that obese people should protect themselves from viral exposure, enhancing protective procedures during SARS-CoV-2 world pandemic. Obese patients [17,69] and severely obese patients [18,23] at younger age [16,30], should be flagged as “high-risk” at the moment of the positive SARS-CoV-2 so that health care providers pay additional attention to this population.

## Figures and Tables

**Figure 1 jcm-09-02511-f001:**
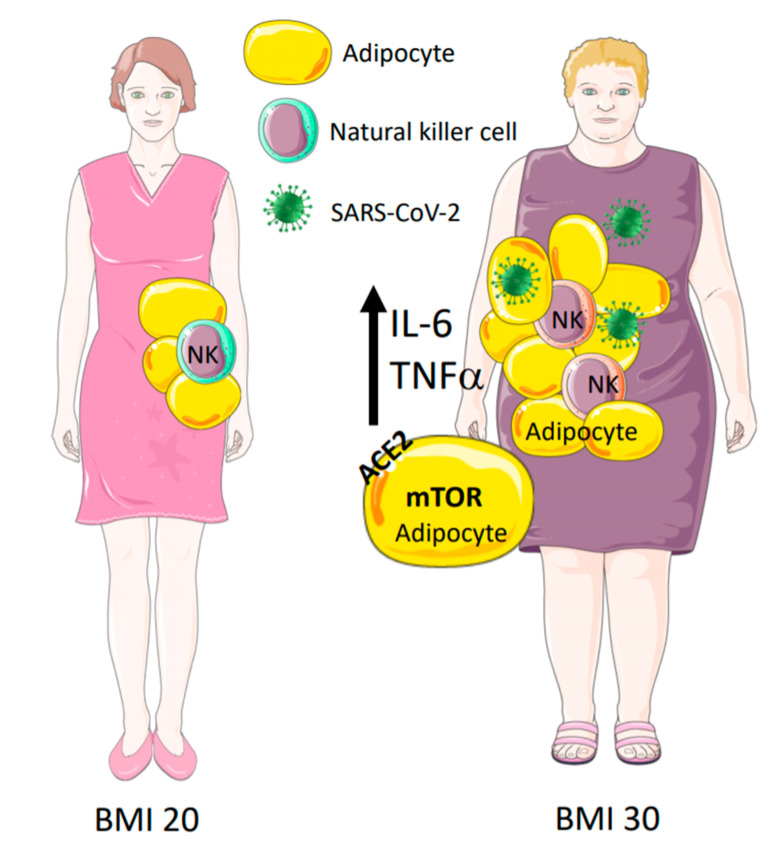
High levels of interleukin 6 (IL-6) and tumor necrosis factor α (TNF-α) are found in obese patients. Natural killer (NK) cells are polarized to non-cytotoxic in obese patients. Adipocytes have on their cell surface angiotensin-converting enzyme-2 (ACE2) and hyperactivation of mammalian target of rapamycin (mTOR) in obese patients, increasing the duration of virus shedding.

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
