# Peer review of "COVID-19 and Obesity: Dangerous Liaisons"

_jcm, 2020, doi:10.3390/jcm9082511_

Round 1

Reviewer 1 Report

Caci et al. have reviewed the association of obesity and COVID-19. They have discussed comorbidities, some alterations at molecular levels, an imbalanced immune response, and dysregulated cytokine network in obese people. They also mentioned the role of ACE2 in the pathogenesis of COVID-19 in obese people. Moreover, they stated the difficulty of positioning obese people by nurses and other health professionals and other problems directly related to extra weight. The article is well-written. I have a few suggestions.

1. regarding obesity and management of patients, the authors would also discuss and cite the article below:

Harris JA, Byhoff E, Perumalswami CR, Langa KM, Wright AA, Griggs JJ. The Relationship of Obesity to Hospice Use and Expenditures: A Cohort Study. Ann Intern Med. 2017;166(6):381-9.

2. Regarding the molecular pathogenesis of obesity, the mTOR pathway, the master regulator of metabolism, is deserved to be mentioned. The pathway is dysregulated in obesity with a potential role in COVID-19. I would recommend the following article:

Bolourian A, Mojtahedi Z. Obesity and COVID-19: The mTOR pathway as a possible culprit. Obes Rev. 2020;10.1111/obr.13084. doi:10.1111/obr.13084

Author Response

  1. regarding obesity and management of patients, the authors would also discuss and cite the article below:

Harris JA, Byhoff E, Perumalswami CR, Langa KM, Wright AA, Griggs JJ. The Relationship of Obesity to Hospice Use and Expenditures: A Cohort Study. Ann Intern Med. 2017;166(6):381-9.

The article by Harris et al. has been included and discussed in the text.

  1. Regarding the molecular pathogenesis of obesity, the mTOR pathway, the master regulator of metabolism, is deserved to be mentioned. The pathway is dysregulated in obesity with a potential role in COVID-19. I would recommend the following article:

Bolourian A, Mojtahedi Z. Obesity and COVID-19: The mTOR pathway as a possible culprit.Obes Rev. 2020;10.1111/obr.13084. doi:10.1111/obr.13084

We have now expanded the molecular pathogenesis of obesity to include mTOR pathway and cited the paper by Bolourian et al. in the text.

Reviewer 2 Report

Dear Editor,

Caci et al. wrote this interesting review article titled “COVID-19 and Obesity: dangerous liaisons” summarizing the evidence supporting that obesity, especially severe obesity, is independently associated with worse outcomes in patients with COVID-19. They present the pathophysiologic mechanisms that explain this association and conclude with some recommendations. I commend the authors for this nicely written review and present my comments below.

INTRODUCTION:

  1. Line 1: The disease is called COVID-19. Would use this term instead of SARS-CoV-2 disease.
  2. Lines 7-8: Would omit this part about smoking. Essentially all the available study are retrospective observational studies that rely on documentation. Smoking status is usually underestimated in these studies. Clinicians do not necessary document smoking status especially in the emergency room or in an admission to the inpatient setting.
  3. Are you certain that these physiologic mechanisms and outcomes apply to overweight people, as well that is BMI 25-29.9)? Or only to obese people (BMI>30) or particularly to severely obese ones (BMI>35)?

EPIDEMIOLOGIC DATA:

  1. Paragraph 1: The association between severe obesity and worse outcomes in COVID-19 has been well-described. Many studies have repeatedly shown this association.
  2. Paragraph 2: Would also mention this study that was one of the first studies to show the independent association of severe obesity with higher in-hospital mortality

Palaiodimos L, Kokkinidis DG, Li W, Karamanis D, Ognibene J, Arora S, Southern WN, Mantzoros CS. Severe obesity, increasing age and male sex are independently associated with worse in-hospital outcomes, and higher in-hospital mortality, in a cohort of patients with COVID-19 in the Bronx, New York. Metabolism. 2020 Jul;108:154262.

  1. Paragraph 2: The study of Richardson et al (reference 16) did not mention outcomes based on BMI, if I recall well. Please let me know if I am wrong. If I am right, I would omit this part.

INVOLVED PATHWAYS:

  1. Very well described. I enjoyed.

HOW OBESITY CAN BE RELATED TO SARS-COV-2 DISEASE PATHOPHYSIOLOGY AND SEVERITY

  1. Paragraph 2, Line 4: Again, I am not aware of research showing that overweight people (BMI 25-29.9) have issues with intubation. I believe obese people do and particularly severely obese patients.
  2. It seems that duplicate information is presented in this section. The pathophysiology has already been presented in the prior section. Should these two sections be merged? Would leave this to the authors to decide.

WHAT ARE THE IMPLICATIONS OF THE FINDINGS?

  1. Should stronger implementation of mitigation strategies focused on severely obese people be applied so they have lower risk for exposure to SARS-CoV-2?
  2. Should severely obese patients be flagged as “high-risk” at the moment of the positive SARS-CoV-2 PCR so that physicians, nurses, and therapists pay additional attention to this population?

Author Response

INTRODUCTION:

  1. Line 1: The disease is called COVID-19. Would use this term instead of SARS-CoV-2 disease.

I think we have only used the term “SARS-CoV-2 disease” only once to introduce the acronym COVID-19. We have now made sure we have used the acronym throughout the text.

  1. Lines 7-8: Would omit this part about smoking. Essentially all the available study are retrospective observational studies that rely on documentation. Smoking status is usually underestimated in these studies. Clinicians do not necessary document smoking status especially in the emergency room or in an admission to the inpatient setting.

The data on smoking status is perhaps one of the most exciting aspect of COVID-19 comorbidities. The smoking findings have been very consistent across multiple countries globally and confirmed by sound meta-analyses (the largest has been cited in the text). Moreover, even population studies are now showing the protective effect of smoking against COVID-19 hospitalization (Israel, et al. Smoking and the risk of COVID-19 in a large observational population study. medRxiv June 1, 2020. doi: https://doi.org/10.1101/2020.06.01.20118877). We agree with the referee that smoking reporting may be inaccurate under the circumstances of a busy COVID ward (as discussed in the reference 7), but - for the same reason - also the recording of body weight (and particularly of BMI) could be fatally flawed. No changes have been introduced in the text, but a reference has been added.

  1. Are you certain that these physiologic mechanisms and outcomes apply to overweight people, as well that is BMI 25-29.9)? Or only to obese people (BMI>30) or particularly to severely obese ones (BMI>35)?

This referee is correct in pointing out that the physiologic mechanisms described in the text apply only to obese patients and not to overweight people. We have amended this in the text.

EPIDEMIOLOGIC DATA:

  1. Paragraph 1: The association between severe obesity and worse outcomes in COVID-19 has been well-described. Many studies have repeatedly shown this association.

This is correct, as presented in the section of our manuscript. We have now populated the existing literature with a few additional key titles (4 new references).

  1. Paragraph 2: Would also mention this study that was one of the first studies to show the independent association of severe obesity with higher in-hospital mortality

Palaiodimos L, Kokkinidis DG, Li W, Karamanis D, Ognibene J, Arora S, Southern WN, Mantzoros CS. Severe obesity, increasing age and male sex are independently associated with worse in-hospital outcomes, and higher in-hospital mortality, in a cohort of patients with COVID-19 in the Bronx, New York. Metabolism. 2020 Jul;108:154262.

Thank you for pointing to this important paper. We have now included and discussed it in the text.

Paragraph 2: The study of Richardson et al (reference 16) did not mention outcomes based on BMI, if I recall well. Please let me know if I am wrong. If I am right, I would omit this part.

The study by Richardson et al. distinguishes COVID-19 outcomes between patients with obesity and morbid obesity. Also, those with diabetes were more likely to have received invasive mechanical ventilation or care in the ICU.

INVOLVED PATHWAYS:

  1. Very well described. I enjoyed.

Thank you for this positive comment. We have now expanded the molecular pathogenesis of obesity to include mTOR pathway.

HOW OBESITY CAN BE RELATED TO SARS-COV-2 DISEASE PATHOPHYSIOLOGY AND SEVERITY

  1. Paragraph 2, Line 4: Again, I am not aware of research showing that overweight people (BMI 25-29.9) have issues with intubation. I believe obese people do and particularly severely obese patients.

This referee is correct in pointing out that issue with intubation specifically apply to obese patients only (and not to overweight people). We have amended this in the text.

  1. It seems that duplicate information is presented in this section. The pathophysiology has already been presented in the prior section. Should these two sections be merged? Would leave this to the authors to decide.

We have amended this in the text.

WHAT ARE THE IMPLICATIONS OF THE FINDINGS?

  1. Should stronger implementation of mitigation strategies focused on severely obese people be applied so they have lower risk for exposure to SARS-CoV-2?

We agree that mitigation strategies should be mainly focused to severely obese people. This has been noted in the text.

  1. Should severely obese patients be flagged as “high-risk” at the moment of the positive SARS-CoV-2 PCR so that physicians, nurses, and therapists pay additional attention to this population?

Thank you for the important suggestion. This is now incorporated in this section.